# Encoding ordered structural complexity to covalent organic frameworks

Lei Wei[1,4], Xinyue Hai[1,4], Tongtong Xu[1], Zidi Wang[1], Wentao Jiang[1], Shan Jiang [1], Qisheng Wang [2], Yue-Biao Zhang [1,3] ✉ & Yingbo Zhao [1,3] ✉

Installing different chemical entities onto crystalline frameworks with well-defined spatial distributions represents a viable approach to achieve ordered and complex synthetic materials. Herein, a covalent organic framework (COF-305) is constructed from tetrakis(4-aminophenyl)methane and 2,3-dimethoxyterephthalaldehyde, which has the largest unit cell and asymmetric unit among known COFs. The ordered complexity of COF-305 is embodied by nine different stereoisomers of its constituents showing specific sequences on topologically equivalent sites, which can be attributed to its building blocks deviating from their intrinsically preferred simple packing geometries in their molecular crystals to adapt to the framework formation. The insight provided by COF-305 supplements the principle of covalent reticular design from the perspective of non-covalent interactions and opens opportunities for pursuing complex chemical sequences in molecular frameworks.

Encoding complexity to ordered matrices is the key strategy that nature uses to program life, where base pairs carrying genetic information are anchored to covalently linked nucleotide scaffolds[1]. Inspired by nature, endowing chemical complexity and sequences to ordered scaffolds also represents an important pursuit for synthetic chemistry[2]. Covalent organic frameworks (COF) are crystalline frameworks constructed by linking molecular building blocks with strong covalent bonds, which provide ideal platforms for encoding structural complexity[3–6]. In this context, the structural complexity of COFs refers to the number of non-equivalent sites that can be used to encode chemical heterogeneity, which is essentially related to the size of the crystallographic asymmetric unit in the COF structure. In reticular chemistry, complexity can be achieved by topological design, where specific connectivity between molecular building blocks can direct them to form complex and ordered structures in mathematically predictable ways[7–9]. For COFs, pursuing new topologies often requires lengthy synthesis of multi-dentate molecular building blocks and faces formidable crystallization challenges associated with the strong covalent linkage. As a result, there have been only a handful of COFs synthesized into single crystals. Structural complexity can also be

achieved by a multivariate approach, where different linkers can occupy equivalent sites in the frameworks to give heterogeneity[10–13]. However, in these cases the linkers only have statistic segregation without well-defined spatial distributions. Consequently, it would be valuable to develop a generalizable strategy that can produce structurally complex single crystalline COFs with simple building blocks and framework topology, which can have ordered spatial distribution of different molecular motifs. We believe the key to developing such a strategy relies on harnessing non-covalent interactions between molecular building blocks.

In COF chemistry, covalent linkages play the dominant structural directing role, which is the chemical basis for topological synthetic design[14]. However, structural dynamics, transformation, and isomorphism of COFs all indicate that there is still plenty of room for structure design through non-covalent interactions, which is associated with different bonding conformations and packing modes of molecular building blocks[9,15–17]. Thus, even if the types of molecular building blocks and their covalent connectivity are pre-determined, there would still be a considerable number of plausible COF structures with different bonding conformations and molecular packing modes

[1]School of Physical Science and Technology, ShanghaiTech University, Shanghai 201210, P. R. China. [2]Shanghai Synchrotron Radiation Facility, Shanghai Advanced Research Institute, Chinese Academy of Sciences, Shanghai 201210, China. [3]Shanghai Key Laboratory of High-Resolution Electron Microscopy, ShanghaiTech University, Shanghai 201210, China. [4]These authors contributed equally: Lei Wei, Xinyue Hai. ✉e-mail: zhangyb@shanghaitech.edu.cn; zhaoyb2@shanghaitech.edu.cn

that span a chemical space available for structural design. To navigate such chemical space and obtain COFs with high structural complexity, non-covalent interactions would need to be the primary consideration to increase the diversity of molecular packing modes, which is correlated to the number of non-equivalent sites in the framework. However, COF building blocks by themselves tend to prefer relatively simple packing modes due to their symmetrical structures[14]. Thus, the key to achieving structural complexity is to find the combination of molecular building blocks so that when they are linked in the predetermined connectivity, the non-covalent interaction between them does not allow them to adapt their intrinsically preferred simple packing modes. These molecular building blocks would then be forced to sample larger configurational space and give higher structural complexity.

In this report, we demonstrate such a strategy in the chemical space of the COF-300 series, which are COFs with **dia** topology formed by linking tetrakis(4-aminophenyl)methane (TAM) and terephthalaldehyde derivatives. As **dia** topology is one of the simplest COF topologies and COF-300 is one of the most studied imine COFs[18], we believe achieving high structural complexity in the COF-300 series would truly show the effectiveness of our design strategy. To carry out the COF design driven by non-covalent interactions, we need to first identify the degree of freedom for non-covalent interactions to play structure-directing roles within the constraint of covalent linkage. For COFs with **dia** topology, such degree of freedom is associated with the

spring-like motions of the diamondoid networks, where the interaction between struts can cause the diamondoid frameworks to stretch to various extents, providing a series of accessible framework configurations that span the chemical space for structural complexity[15–18]. Such degree of freedom in **dia** COF is, in a way analogy to that of DNA molecules, where hydrogen bonding between base pairs can assemble the polynucleotide chains into double helices while maintaining the covalent bond between nucleotides (Fig. 1a). Subsequently, proper molecular building blocks need to be selected to harness the degree of freedom in **dia** COFs and achieve structural complexity. In this regard, supramolecular chemistry provides an essential toolbox to identify the molecular motifs where the non-covalent interactions between them can play structural directing roles. It is worth noting that supramolecular chemistry considerations such as molecular pre-assembly have been known to help increase the crystallinity of 2D COFs[19–22]. For the known examples of the COF-300 series, the TAM molecules strongly favor a simple and robust herringbone packing that can be viewed as supramolecular synthon responsible for their relatively simple structure[23]. Thus, to pursue structural complexity, the TAM packing would need to be altered by introducing supramolecular interactions associated with the aldehyde building blocks.

With the above-mentioned design strategy, we choose 2,3-dimethoxyterephthalaldehyde (DMPA) as the aldehyde linker to link with TAM and synthesize a structurally complexed COF-305 with simple dia topology, which is obtained as large single crystals. COF-305 has a well-

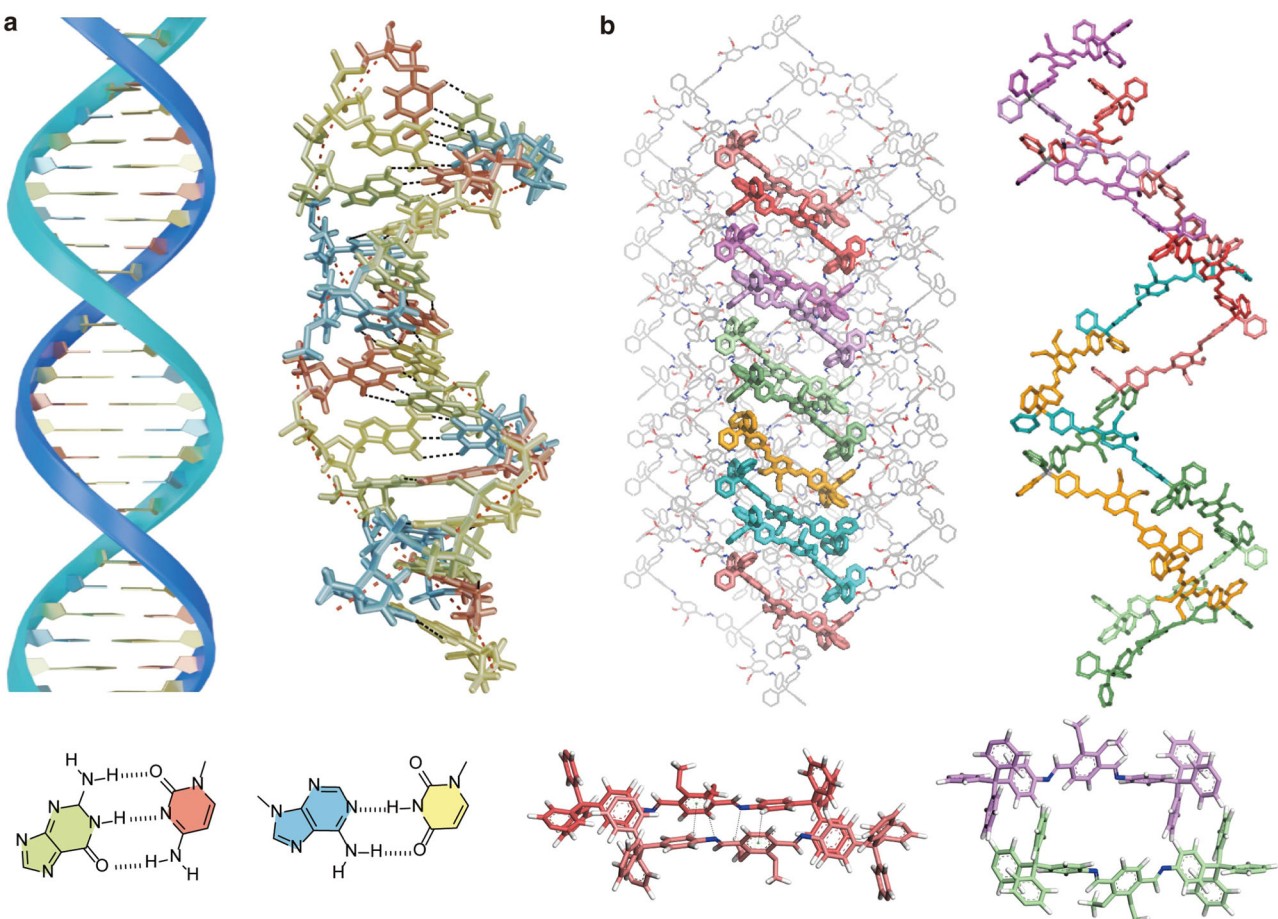

**Fig. 1 | Non-covalent interactions playing structure-directing roles for covalently linked scaffolds with complex structures. a** Leveraging the flexibility of the covalently bonded polynucleotide chain, the hydrogen bonding between base pairs can assemble them into double helices. **b** The structural flexibility of COFs with **dia** topology can be readily seen from their $4_1$-helix chains, which provides the degree of freedom for non-covalent interactions to assemble adjacent struts into complementary pairs and give rise to ordered complexity. In COF-305, nine stereoisomers of its struts, labeled with different colors, show well-defined spatial distributions on the fivefold interpenetrated diamondoid framework.

defined spatial distribution of linker stereoisomers giving rise to specific chemical sequences along certain crystallographic axis (Fig. 1b), with nine crystallographically nonequivalent DMPA and 4.5 TAM with different conformations displaying well-defined sequences on topologic equivalent sites. The cell volume of COF-305 is over 130,000 cubic-angstrom, enclosing 72 TAM and 144 DMPA motif, which is more than one order of magnitude larger than COF-300[18]. The structural motifs linking the central carbon atom of two TAMs appear as configurational complementary pairs, similar to supramolecular recognition between DNA double helix strands (Fig. 1). We also synthesized single crystals of another COF with **dia** topology from TAM and 2,3-dihydroxyterephthalaldehyde (DHPA), termed COF-304, which has structural complexity between COF-300 and COF-305 and provide important structural hints to rationalize the structure of COF-305 (Supplementary Note 1, Supplementary Fig. 1). By comparing the crystal structure of COF-300, COF-304, COF-305, and their molecular model compounds, we conclude that the structural complexity of COF-305 originates from the TAM motif deviating from its most preferred herringbone packing to accommodate the face-to-face packing of the DMPA. From a supramolecular perspective, the herringbone packing of TAM and face-to-face packing of DMPA compete and compromise with each other within the constraints of diamondoid framework connectivity, which results in a substantially increased diversity of building block conformations that eventually lead to structural complexity. From such observation, we further postulate a COF structural design principle from the perspective of non-covalent interactions, which is complementary to the classical principle of covalent reticular design: COF structures can be viewed as covalently stitching pre-assembled molecular building blocks, which have molecular packing geometries both intrinsically favored by the molecular building blocks and can accommodate the COF connectivity. We believe although such a postulate is deduced from COFs with complexity originating from stereo-isomorphism, it provides a generalizable guideline for designing highly complex frameworks with sequenced chemical entities, which represent a viable path to pursue the ultimate limit of chemical self-assembly and emergent sequences in synthetic materials.

## Results and discussions
### The structural complexity of COF-305

COF-305 was synthesized by reacting 20 mg TAM and 17 mg DMTA in 1.25 mL 1,4-dioxane with the presence of 15 M acetic acid (HOAc, 200 µL) and aniline (40 µL) at room temperature. After 4 days, yellowish polyhedral crystals of COF-305 in size of 40 µm were obtained and analyzed by single-crystal X-ray diffraction giving beyond 1 Å resolution. COF-305 crystallized in the space group *Fdd*2 (No. 43) with unit-cell parameters of $a = 47.077(9)$ Å, $b = 67.629(14)$ Å, $c = 42.547(9)$ Å, $\alpha = \beta = \gamma = 90°$, and $V = 135460(47)$ Å$^3$ (Fig. 2a, Supplementary Figs. 2–7, Supplementary Table 1 and 2). COF-305 is also characterized by solid-state $^{13}$C nuclear magnetic resonance, Fourier-transform infrared, thermal gravimetric analyses, and gas adsorption measurements (Supplementary Figs. 8–12). We also synthesized the COF-305

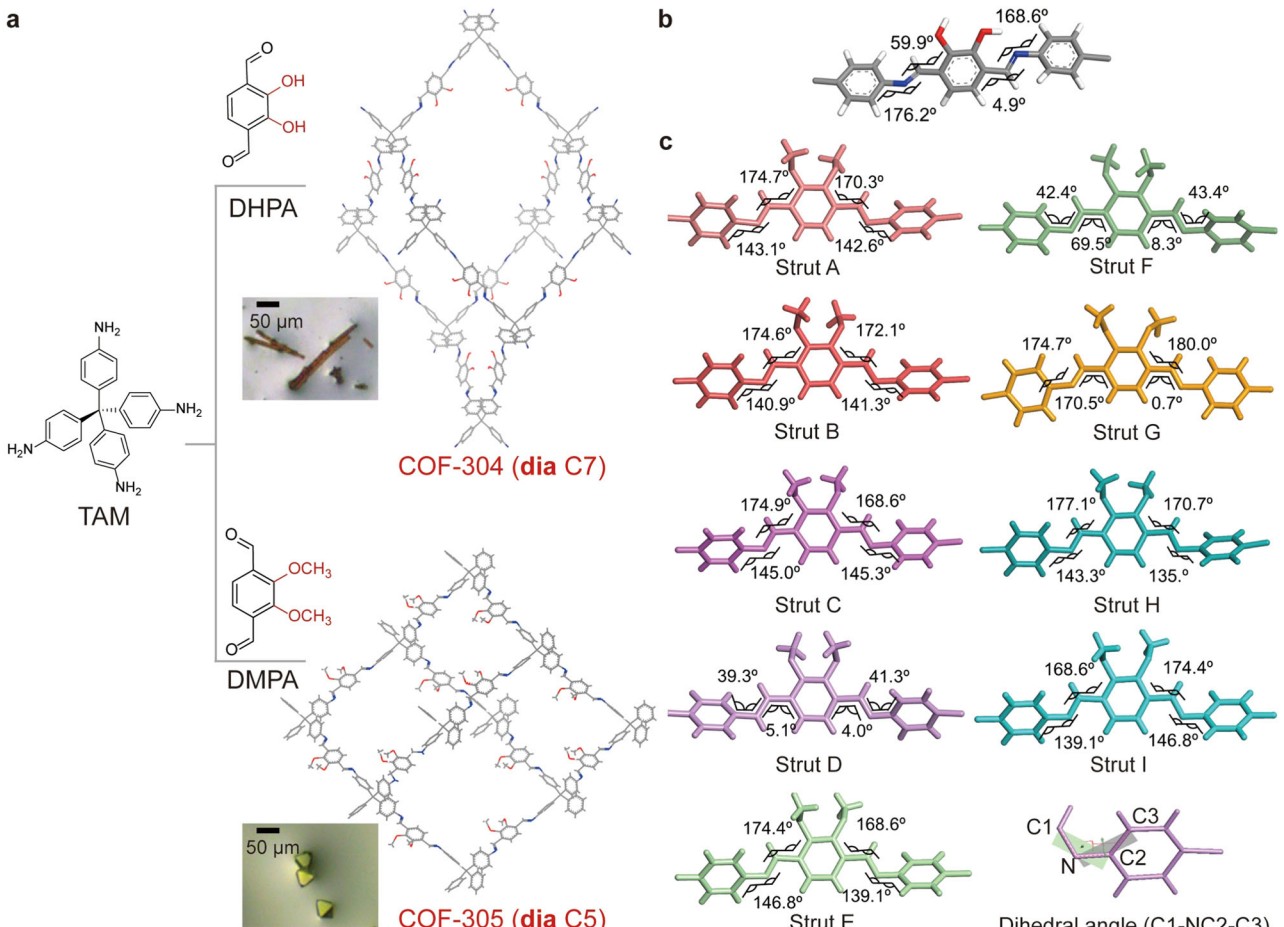

**Fig. 2 | Basic structural motifs of COF-304 and 305. a** Constructed with TAM and dialdehyde building blocks, COF-304 and 305 have diamondoid framework that can be viewed as Ph-N = C-Ph(OX)2-C = N-Ph (X = H for 304, CH3 for 305) struts linked with carbon atom nodes. **b** COF-304 has one type of linking strut, consistent with its symmetrical and simple structure. **c** COF-305 has nine different struts, which are stereoisomers. The dihedral angles between the imine bond (HC = N) plane and the phenyl ring on each strut are labeled.

High-quality single crystals of COF-304 were crystallized by reacting 2,3-dihydroxyterephthalaldehyde (DHPA) and TAM in a similar condition, having space group of $I$-4 (No. 82) and unit cell parameters $a = b = 26.210(6)$ Å, $c = 7.582(3)$ Å, $V = 5208(3)$ Å$^3$, similar to COF-300 (Fig. 2a, Supplementary Figs. 13 and 14, and Supplementary Table 3). The theoretical pore volume of COF-305 and COF-304 are calculated using a probe radius of 1 Å, which are 1.06 and 0.512 cm$^3$ g$^{-1}$, respectively. The pore volume of COF-305 is measured by CO$_2$ uptake at 195 K to be 0.91 cm$^3$ g$^{-1}$ (Supplementary Fig. 12), which is largely consistent with the theoretical result considering the framework deformation during the activation process.

To systematically describe and compare the structure of COF-304 and COF-305, we analyze their structures at three levels, which are struts, diamondoid nets and interpenetration. Here we define the strut as the -Ph-N = C-Ph(X)$_2$-C = N-Ph- (X = H for COF-300, OH for COF-304, OCH$_3$ for COF-305) motif, which is the most basic structural unit. These struts would then link up carbon nodes to form diamondoid nets[24], which can be described by the corresponding primitive interpenetration cells (PIC [25]) through translational symmetry operations. The diamondoid nets would then interpenetrate into the voids of each other to give the COF structure, which can be specified by the degree and direction of interpenetration.

For COF-300, half of the -Ph-N = C-Ph-C = N-Ph- strut constitutes its crystallographic asymmetric unit, and the diamondoid framework can be described by a simple adamantane cage as the PIC. For COF-304, the -Ph-N = C-Ph(OH)$_2$-C = N-Ph- strut constitutes its crystallographic asymmetric unit (Fig. 2b and Supplementary Fig. 14), and its PIC is also an adamantane cage (Supplementary Fig. 15). Both COF-300 and COF-304 have 7-fold interpenetration with the vector of [0 0 1], which means each of the interpenetrating networks are offset on the direction of $c$-axis by one-unit cell length (Supplementary Fig. 16).

The structure of COF-305, however, is much more complex. The COF has nine different Ph-N = C-Ph(OMe)$_2$-C = N-Ph struts in its asymmetric unit (Fig. 2c, Supplementary Fig. 3 and 17, Supplementary Table 4), which are coded from A to I with different colors. These struts are all in "C" configurations, which is likely due to the steric hindrance effect of the methoxy group. The nine struts are stereoisomers and have different length ranging from 18.18 to 18.45 Å (Supplementary Fig. 18), and their differences in dihedral angles between the imine bond and the benzene rings are shown in Fig. 2c. The PIC of COF-305 is composed of a column of nine corner sharing adamantane cages with one additional strut links to the node of each adamantane cage (Supplementary Fig. 19). To analyze the structure of PIC more clearly, we only show the adamantane cage without the peripheral struts in Fig. 3a for simplicity. The nine adamantane cages in PIC are coded I to IX according to their sequence along the $b$-axis. The cages can also be labeled by the struts they are made from. For example, cage IV can be marked as $F_2G_4H_4I_2$, indicating that it is composed of two struts F, four struts G, four struts H, and two struts I. In this notation, four pairs of cages have the same compositions, and they could be obtained by symmetry operation (rotating 180° along the $c$-axis followed by a translation) (Supplementary Fig. 20). Thus, the nine struts produce five types of non-equivalent cages that constitute the PIC of COF-305 (Supplementary Fig. 21). It is worth noting that the adamantane cages are more squashed along the stacking direction in COF-305 than COF-300 and 304, indicating the emergence of ordered complexity indeed leverages the complexity of the covalently bonded **dia** network (Supplementary Fig. 22). The COF-305 is 5-fold interpenetrated, where five symmetry-equivalent diamondoid networks are related by translation vector [0 1/2 1/2] (Fig. 3b and Supplementary Fig. 23). The diagonal translation vector of the interpenetration result in different adamantane cages to be placed in the vicinity of each other, giving rise to specific sequences of the nine struts along the $b$-axis, which is the key structural feature of COF-305 showing its structural complexity.

Viewing along $b$-axis, the struts show only two types of well-defined sequence, ABCDEFGHI and its reverse, depending on the column of struts being analyzed (Fig. 3c and Supplementary Fig. 24). In one column, the struts form four pairs, AB, CD, EF, and HI, with struts in each pair having closer distances and curved in opposite directions (Fig. 1b and Supplementary Fig. 25). The other strut G (labeled orange) is unpaired and more distanced from the adjacent F and H strut. For simplicity, the structural units in the same pair are coded by the same type of color. The pairing of struts opens void spaces between different pairs, which form channels perpendicular to the packing direction of TAM with the size of 7.3–8.0 Å. These channels are perpendicular to the one-dimensional channel along $b$-axis, which gives COF-305 a three-dimensional porous structure (Supplementary Fig. 26). In contrast, COF-304, same as COF-300 and other **dia** COFs, only have one-dimensional channels along the direction of TAM packing (Supplementary Fig. 27). In summary, COF-305, made from simple building blocks and have simple **dia** topology, shows structural complexity fundamentally beyond any of the known COFs and the emergence of sequenced linker conformers. The well-defined sequence of these linker conformers on topologically equivalent sites represents a new strategy of COF complexity design, which we would unravel by analyzing the packing geometry and non-covalent interactions associated with DMPA and TAM motif.

## The packing geometry of molecular building blocks in COFs and model compounds

The molecular building block packing geometry in COF-305, along with that of COF-300 and COF-304, is then analyzed to shed light on the origin of its ordered complexity. The TAM building block is known to favor a herringbone packing mode (Supplementary Table 5 and Supplementary Fig. 28), which is also a classic synthon in supramolecular chemistry[23]. Herringbone packing is observed for TAM, tetraphenyl methane (TPM) (Supplementary Table 6 and Supplementary Fig. 29), COF-300, and COF-304, but not for COF-305. The key feature of the herringbone packing is the 67° dihedral angles between the interacting benzene rings (Fig. 4a). In COF-300, this 67° dihedral angle is maintained. For COF-304, which has lower symmetry, two dihedral angles of 67° and 62° are found, and the deviation from 67° is associated with the in-plane hydrogen bonding and steric effect involving the hydroxyl groups (Supplementary Fig. 30). For COF-305, the majority of TAMs are packed in a zigzag mode, which is distinctively different from herringbone packing and has the dihedral angles below 40°. (Fig. 4b, c, and Supplementary Fig. 31). There is another packing mode of TAMs in COF-305 having two pairs of C-H···C interactions instead of four, which we will name as displaced packing and discuss later (Fig. 4b and Supplementary Fig. 32).

For the packing geometry of aldehyde building blocks, single crystals of the model compounds made by reacting aniline with terephthalic aldehyde, DMPA, and DHPA are analyzed, which we refer to as M-300, M-304, and M-305, respectively (Supplementary Figs. 33–37 and Supplementary Tables 7–11). It is found that while M-300 and M-304 have phenyl rings in "side-by-side" packing, M-305 has mostly face-to-face packing. In all three COFs, the aldehyde building blocks largely show similar packing modes as their corresponding molecular model compound, with different degrees of local deviations (Fig. 4d–f). Thus, comparing the structure of COF-305 and model compounds, it can be found that in COF-305, the TAMs made more compromises in their packing mode, shifting from intrinsically preferred herringbone packing to zigzag and displaced packing, whereas the DMPA makes only local compromises, maintaining the "face-to-face" packing. However, such observation does not necessarily mean that the DMPA "pair" are energetically stronger synthon than the TAM in herringbone packing, as all these compromises and completions take place in the context and constraint of the covalent connectivity of the diamondoid network. In the next section, we will analyze the molecular packing and

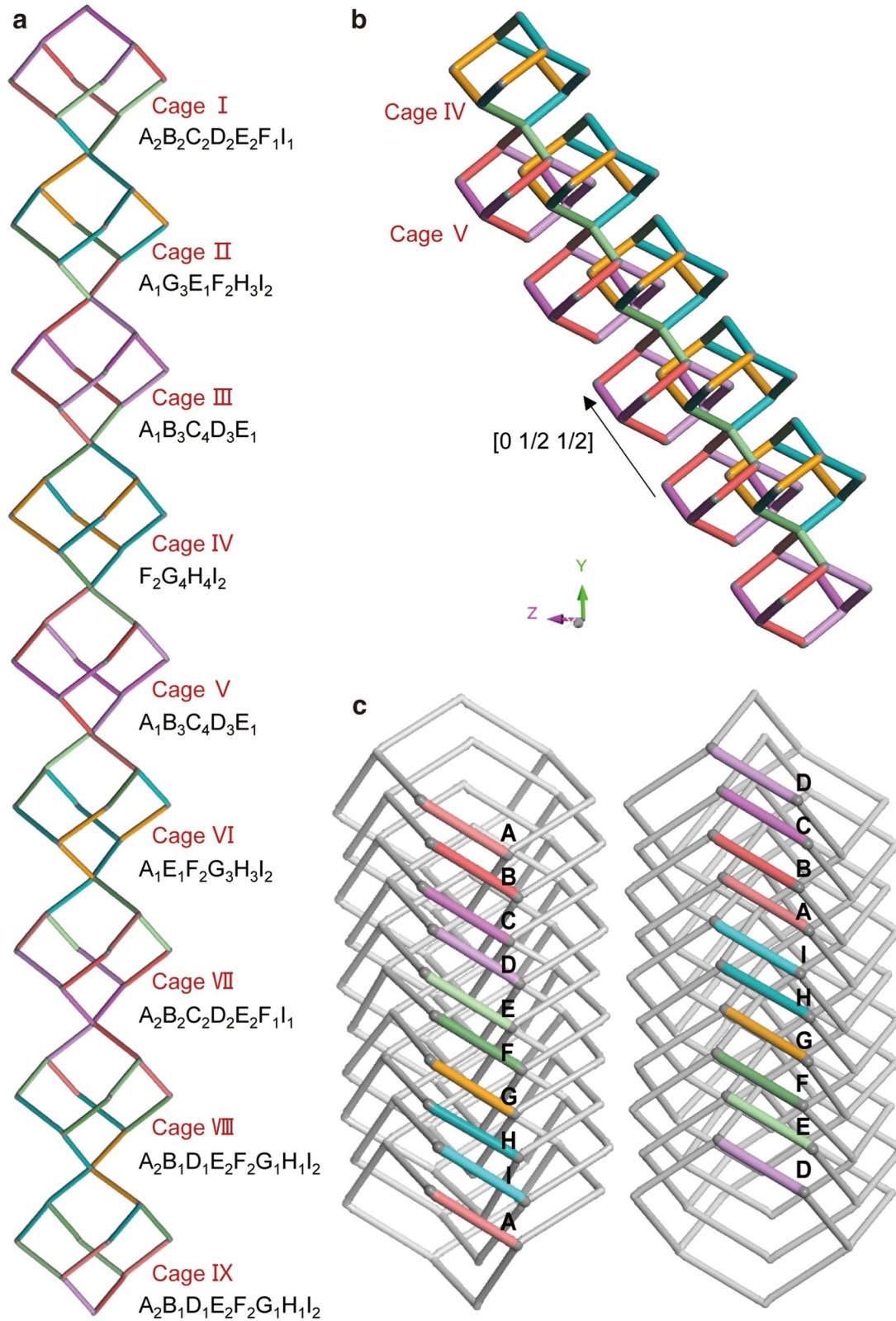

**Fig. 3 | The ordered complexity of COF-305 and sequences of struts. a** The primitive interpenetration cell (PIC) of COF-305 consists of nine adamantane cages in a column, each incorporating twelve struts. The diamondoid network of COF-305 can then be generated from the PIC through translational symmetry operations. **b** Five diamondoid frameworks offset by translation vector [0 1/2 1/2] interpenetrate to give COF-305. **c** The key feature of ordered complexity in COF-305 is that the nine different struts show specific sequences along the *b*-axis.

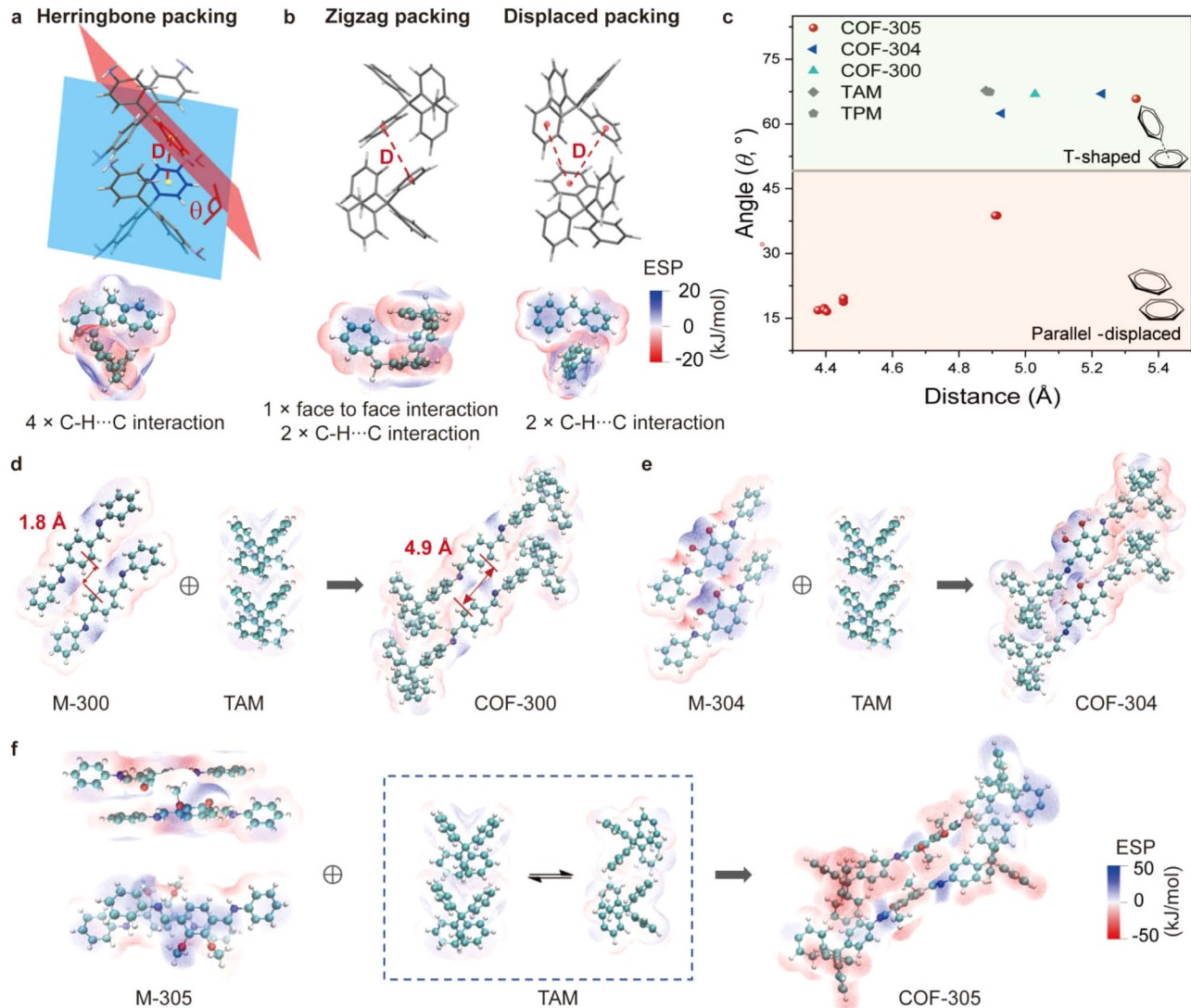

**Fig. 4 | Molecular building block packing geometry adaptation observed in COF-305 series. a** The herringbone packing of TAM motif is characterized by four pairs of CH···C interactions shown in the Hirshfeld surface analysis, where the phenyl rings have a dihedral angle around 62°. **b** In COF-305, the TAM shows zigzag packing and displaced packing mode. **c** The TAM packing modes in COF-300, 304, 305, and model compounds can be quantitatively described by the phenyl ring dihedral angle and distances, which shows the TAM packing in COF-305 is distinctively different from the others. **d** The structure of COF-300 can be viewed as combining terephthalic aldehyde and TAM in each of their preferred packing modes shown in their molecular compounds with only local changes to the displacement value between the aldehyde motif. **e** The structure of COF-304 can also be viewed as combining the DHPA and TAM in each of their preferred packing geometry while changing the relative orientation of the two imine bonds from a "C" to an "S" shape. **f** For COF-305, both TAM and DMPA deviate from their preferred packing mode to adapt to framework formation, giving rise to ordered complexity.

covalent linkage together in one picture and provide a new vision of COF structures from the perspective of molecular assembly that rationalizes the emergent of ordered complexity.

## Building block packing geometry adaptation and COF structural complexity

For COF-300, both the aldehyde and TAM have minimal deviations from their intrinsically preferred packing, which is evident by the small structural differences between COF-300, TAM, and M-300 shown above. The most prominent deviation is the phenyl rings of the terephthalic aldehyde in COF-300 have a slightly larger offset compared to M-300, presumably to accommodate both the preferred herringbone packing of TAM and the imine bonds (Fig. 4d). Thus, the structure of COF-300 can almost be viewed as directly stitching the prepacked TAM and aldehyde motif through imine bonds. For COF-304, the structural deviations from packing geometry observed in the model compounds are slightly more pronounced due to the hydrogen

bonding and steric associated with the OH group, with the dihedral angle of TAM shifted from 67° to 62° and the relative orientation of two imine bonds change from "C" to "S" shape (Fig. 4e, Supplementary Fig. 38). Comparing COF-300 and 304, it already shows that introducing additional interactions between molecular building blocks can perturb the intrinsically preferred packing more severely and lead to more deviations from these packing modes after COF formation.

For COF-305, stitching TAM in herringbone packing with DMPA would not give a feasible COF structure due to the steric hindrance (Supplementary Fig. 39), which requires the TAM to adapt to distinctively different zigzag and displaced packing modes. To verify the effect of framework formation on TAM packing, we synthesized another model compound M-305-2 (Supplementary Table 11, Supplementary Fig. 40, and Supplementary Fig. 41) from TAM and 2,3-dimethoxybenzaldehyde, where both TAM and dimethoxybenzaldehyde show their intrinsically preferred herringbone and face-to-face packing without the restriction of imine-linked

framework. On the other hand, DMPA in COF-305 also shows local slide displacement in three dimensions to accommodate the imine linkage with TAM without dramatically changing the highly preferred face-to-face packing geometry (Supplementary Table 12). In summary, the ordered complexity in COF-305 is the result of both molecular building blocks deviating from their intrinsically favored simple packing modes to accommodate each other in a covalently linked framework.

Based on the observation above, we could propose three scenarios for COF structure from the perspective of building block packing, which are direct stitching, local adaptation, and global adaptation: (a) the molecular building blocks can form the COF with minimal deviation from their preferred packing geometry, as if the pre-packed building blocks are simply stitched by covalent linkages (e.g., COF-300); (b) the molecular building blocks need to make local adaptations to accommodate the inter-molecular interactions involved in the framework formation (e.g., COF-304); (c) the molecular building blocks need to make global adaptations on their packing (e.g., COF-305), which could give a series of configurations with similar

energy and result in structural complexity (Supplementary Fig. 31e). The global adaptation scenario embodied by COF-305 is then quantitatively analyzed, where the rich library of stereoisomers readily shows a series of configurations.

Energy calculation of the nine pairs of interacting struts in COF-305 quantitatively shows how molecular building blocks make adaptations in their packing geometry to accommodate each other in the framework (Fig. 5, and Supplementary Table 12). The energy of the interacting struts is mainly affected by two factors: the packing mode of TAM and the lateral offset value of DMPA. For TAM, calculation shows the zigzag mode is energetically more favorable than displaced packing: the GH struts with displaced packing on both sides show the highest energy, and the FG and HI struts having displaced packing on one side show the next highest energy. For the other six pairs with all zigzag TAM packing, the offset value of DMPA determines their energy. M-305 represents the intrinsically favored packing of DMPA, and more deviation from this structure gives higher energy, which is evident by the correlation between energy and the lateral offset value

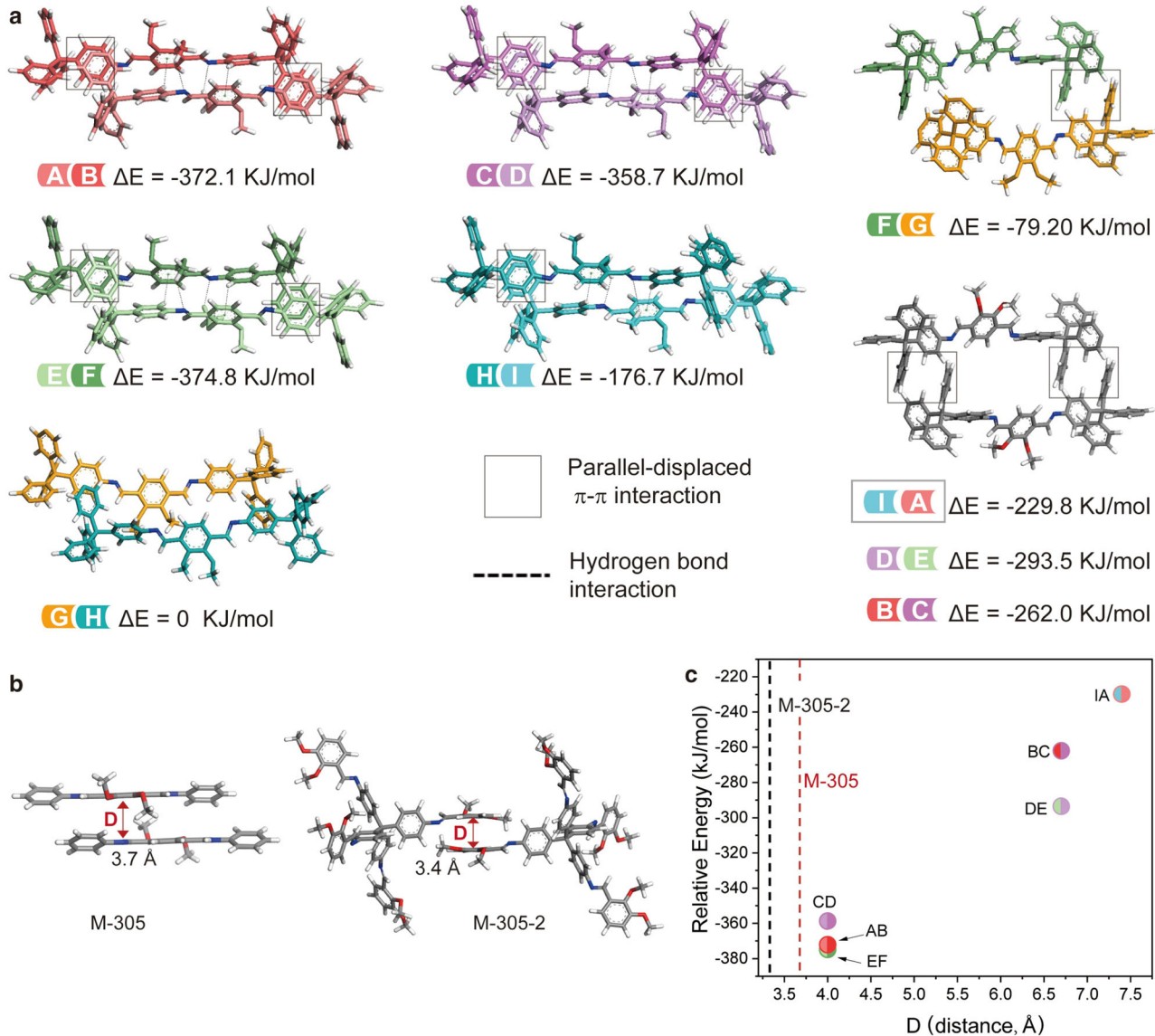

**Fig. 5 | Quantitative analysis of building block packing geometry adaptation. a** The single-point energies of adjacent struts are calculated with the energy of the weakest interacting GH pair as a reference point. The prominent non-covalent interactions are labeled, showing both the interaction between adjacent TAM and DMPA influence the energy. **b** The intrinsically preferred packing mode of the

DMPA motif is shown in the model compound M-305 and M-305-2, with the lateral offset distance D being around 3.5 Å. **c** The extent of DMPA deviating from such preferred packing geometry to adapt to framework formation in COF-305 can be quantified by this lateral offset, which is correlated to the relative energy of struts with the same TAM packing.

between DMPA (Fig. 5c). Thus, it is clear that both TAM and DMPA deviates from their intrinsically preferred packing mode with low energy to form the **dia** COF, which in return gives structural complexity.

We believe the scientific implications of our findings on COF-304 and 305 are not limited to COF chemistry and could be generalized to other framework materials designed by reticular chemistry. In general, reticular design employs strong and directional interactions to link up molecular building blocks in mathematically predictable ways with well-defined connectivity. These directional and dominant interactions can be viewed as the primary structure-directing interaction for the corresponding frameworks: covalent bonds for COFs, metal-to-ligand bonds for MOFs, hydrogen bonds for hydrogen-bonded frameworks (HOFs), and so on[26,27]. For each framework category, the primary interaction would almost always leave a degree of freedom for weaker and less directional interactions to play secondary structural directing roles and give structural diversity, complexity, and tunability. For example, such structural tunability through secondary structural directing interactions can also be found in MOF chemistry, where interpenetration and conformational isomers can be obtained by tuning non-covalent interactions, such as the solvent templating effect[28]. Our findings in COF-305 have shown that systematically understanding and tuning secondary structural directing interactions could lead to profoundly different structures, even for framework materials that seem to be very simple from the perspective of primary interactions and topological connectivity.

In summary, we have synthesized COF-305 from simple building blocks with dia topology that shows high complexity embodied by large unit cells and well-defined spatial distribution of nine different stereoisomers of the molecular building units on topologically equivalent sites, which provide a fundamentally new strategy to encode chemical complexity to COFs. Comparing the structure of COF-305, 304, 300, and model compounds reveals the adaptation of molecular building blocks packing geometry in COF formation is an essential factor determining the COF structure. Such observation provides a unique vision of COF chemistry from the perspective of non-covalent interactions that supplements the classic reticular chemistry centralizing on covalent linkages. Such vision can be generalized to other framework materials, including MOFs, HOFs, and even molecular compounds, as in these materials, a primary interaction provides dominant structural directing roles but also leaves a degree of freedom for weaker, secondary interactions to endow structural complexity. These findings represent a new interdisciplinary frontier of reticular chemistry and supramolecular chemistry and would open opportunities for the design of complex and functional framework materials.

## Methods

### Materials
2,3-dihydroxyterephthalaldehyde (DHPA, purity ≥ 98%), 2,3-dimethoxyterephthalaldehyde (DMPA, purity ≥ 98%), 4",4"'-methanetetrayltetraaniline (TAM, purity ≥ 99%) were purchased from Jilin Chinese Academy of Sciences-Yanshen Technology Co., Ltd. Terephthalaldehyde (TPA, purity ≥ 98%) was purchased from Shanghai Aladdin Biochemical Technology Co. Super dry 1,4-dioxane (purity ≥ 99.5%) and aniline (purity ≥ 99.5%) were purchased from J&K Scientific Ltd. 1,4-dioxane (AR, purity ≥ 99.5%) and n-butanol (AR) were purchased from General-reagent Co., Ltd. Purification of TAM is needed prior to COF synthesis.

### Instrumentation
Single-crystal X-ray diffraction (SXRD) data of COF-304 and TPM was collected at Shanghai Synchrotron Radiation Facility (SSRF) Beamline BL17B1 station ($\lambda = 0.71073$ Å) equipped with a marXperts MX300 CCD detector. SXRD of COF-305 was collected at SSRF Beamline BL10U2 station ($\lambda = 0.6887$ Å) equipped with a marXperts MX300 CCD detector. All the datasets of single-crystal were collected at 100 K, and the data were accordingly processed with CrysAlisPro (version 171.37.35), HKL2000[29], and APEXII[30] software packages, depending on the instrument setup. SXRD data of TAM, DMPA, M-300, M-304, M-305, and M-305-2 was collected on a Bruker D8 Venture Photon III diffractometer using Ga K$\alpha$ ($\lambda = 1.34138$ Å) X-ray source. The specimen was cooled to 150 K using a cryo-stream 800 plus chilled by liquid nitrogen. SXRD data of DHPA was collected on a Bruker D8 Venture diffractometer equipped with a fine-focus Cu target X-ray tube operated at 40 W power (50 kV, 1 mA) with radiation of $\lambda = 1.5418$ Å. The specimen was cooled to 150 K using a cryo-stream 800 plus chilled by liquid nitrogen. The structure analysis and refinement were carried out using the SHELX algorithms in Olex2[31,32]. Solvent masking was applied during structure refinement[33]. All the non-hydrogen atoms were refined anisotropically, all the hydrogen atoms were added geometrically and refined in the riding model. Crystal data and details of the structure refinement for single crystals are given in Supplementary Tables and the attached CIFs. Mercury and Diamond's software were used for structural visualization.

Powder X-ray diffraction (PXRD) patterns of COF samples were collected on a Rigaku D8 Advance diffractometer equipped with a Cu target X-ray tube operated at 1600 W power (40 kV, 40 mA). The samples were measured with a step size of 0.02° and a scan time of 0.2 s per step. The Fourier-transform infrared (FT-IR) COF samples were recorded on neat samples in the range of 500–4000 cm$^{-1}$ on a PerkinElmer FT-IR spectrometer equipped with a single reflection diamond ATR module. The solid-state 13 C nuclear magnetic resonance (13 C SSNMR) of COF samples was recorded on a Bruker ADVANCE 400 MHz Solid NMR spectrometer with cross-polarization magic angle-spinning (CP/MAS) and 4.0-mm double-resonance MAS probe. The samples were loaded in the zirconia rotor with a sample spinning rate of 13.0 kHz. The thermal gravimetric analyses (TGA) were carried out using a TAQ50 TGA analyzer from 25 to 800 °C under an N2 atmosphere with a temperature ramping rate of 5 °C/min. Low-pressure gas adsorption isotherm was measured volumetrically using a Quantachrome iQ (N$_2$, CO$_2$). Liquid nitrogen and dry ice-methanol baths were used for temperature control at 77 K and 195 K, respectively.

### DFT calculations of COF-305 fragment pairs
For accurate energetic comparisons between the COF-305 fragment pairs, DFT calculations were conducted on each pair. Due to the typically inaccurate positions of H atoms obtained by single crystal X-ray diffraction[34] it is necessary to perform geometry optimizations before single-point energy calculations. Meanwhile, to investigate the differences among COF fragments in the crystal structure, we performed constrained geometry optimizations. The starting atomic coordinates were taken from single crystal diffraction. The coordinates of hydrogen atoms were optimized using Gaussian09[35] with the B3LYP functional[36] combined with the def2-SVP basis set[37] and Grimme D3 dispersion corrections[38]. Then, single-point energy calculations were performed on DFT-optimized structures using the B3LYP functional, the def2-TZVP basis set[37,39], and Grimme D3 dispersion corrections.

### Synthesis of COF-304
DHPA (15 mg, 0.090 mmol) was dissolved in 0.5 mL of 1,4-dioxane, and 60 µL of aniline was added. The mixture was transferred to an NMR tube, and then 0.2 mL of aqueous acetic acid (15 M) was added. After homogenously mixed, a red-crystalline precipitate was observed, and the solution of TAM (20 mg, 0.053 mmol) dissolved in 0.75 mL 1,4-dioxane was added carefully. The reaction was placed at room temperature stand still for three weeks, the dark red-colored crystals slowly crystallized out. The as-synthesized single crystals were isolated and then exchanged by n-butanol three times for SXRD measurement.

A rod-shaped crystal ($10 \times 10 \times 60\ \mu m^3$) of COF-304 was selected for SXRD measurement.

## Synthesis of COF-305
DMPA (17 mg, 0.088 mmol) was dissolved in 0,5 mL of 1,4-dioxane, and then 40 μL of aniline and 200 μL aqueous acetic acid (15 M) were added to a 5 mL vial. After homogeneously mixing the solution, the solution of TAM (20 mg, 0.053 mmol) dissolved in 0.75 mL 1,4-dioxane was added carefully. The reaction was placed standstill at room temperature. The yellow-colored crystals of about 30 μm were crystallized out after 2 days, and the crystal size could reach about 40 μm in 4 days. The as-synthesized single crystals were isolated and then exchanged by *n*-butanol three times for SXRD measurement.

## Data availability
The data that support the findings of this study are available from the corresponding authors upon request. The crystal structure of COF-305, COF-304, and corresponding small molecules generated in this study have been deposited in the Cambridge Crystallographic Data Center under access code CCDC#2292628-2292637. Source data are available in this paper. All data are available in the main text or the Supplementary Materials. Source data are provided in this paper.

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

## Acknowledgements
The authors thank beamline-BL17B1 and BL10U2 of the Shanghai Synchrotron Radiation Facility for providing the beamtime. Dr N. Yu at ShanghaiTech University for assistance during SXRD data collection. Ms L. Long at ShanghaiTech University for assistance during gas adsorption measurements. Y.Z. acknowledge support from the Science and Technology Commission of Shanghai Municipality (22QC1401500). Y.-B.Z. acknowledge financial support from the National Natural Science Foundation of China (grants 22271189 and 21522105), the Science and Technology Commission of Shanghai Municipality (grants 21XD1402300, 21JC1401700, and 21DZ2260400), and the Double First-Class Initiative Fund of ShanghaiTech University.

## Author contributions
Y.Z., Y.B.Z., and L.W. initiated and led the research project. L.W. and X.H. conducted the synthesis and crystal growth. L.W., T.X, Q.W., and Y.B.Z. conducted the single-crystal X-ray diffraction measurement and analysis. Z.W. and S.J. carried out the theoretical calculation. L.W. and W.J. carried out the electrostatic potential analysis. Y.Z. and L.W. organized the result and wrote the paper with discussion and revision of all authors.

## Competing interests
The authors declare no competing interests.
