## [Peer Review File · Nature Communications]

REVIEWER COMMENTS

Reviewer #1 (Remarks to the Author):

The manuscript reports the exceptional complexity in COFs, resembling the hierarchical structure of DNAs. The unprecedented diversity of conformations of COFs observed by X-ray single-crystal structures with atomic precision have been elaborated in great details in both molecular geometry and energy aspects. The molecular insight revealed here is very inspiring for the delicate design of COF structures with the consideration of linker-linker interactions for controlling their spatially arrangement of building blocks. Such results are out of expectation but can be well rationale, manifesting the power of serendipity in reticular chemistry. The experiments have been well-conducted and the manuscript is well-written. Therefore, I recommend it for the publication in Nature Communications after minor revision. Several comments are provided as follows:

1. The authors have illustrated the complementary pairs of the DMPA motifs. It will be very interesting to know whether the formation of such complexity arise from the pre-assembly of monomer in reactant.
2. The interpenetration number of COF-305 is only 5-fold, while the COF-304 has 7-fold similar in the case of COF-300. Is it possible to explain how the methoxyl group make a impact in the interpenetration and the translational direction?
3. As we all known that organic solvents can play as structural directing agents in the formation of porous crystals. Can the author elaborate the impact of organic solvents using in COF-305 leading the such high level of complexity?

Reviewer #2 (Remarks to the Author):

The manuscript (MS) "Encoding Ordered Complexity to Covalent Organic Frameworks" by Zhao and coworkers describes the synthesis, characterization, and structural analysis of diamond COFs (COF-304, COF-305) in detail. From crystallographic point of view, this analysis is valuable. However, the MS lacks of novelty and does not provide inspired insights into the design of new structure types in COF chemistry (please see the below concerns and questions). Therefore, I recommend the editor not publish this work in Nature Communications. A more specific journal related to crystallography is suitable for this MS.

1. Topology not only concerns covalent bonds but also relates various bonds as topology describes the connectivity (see Cryst. Growth Des. 2014, 14, 4, 1938–1949 and Cryst. Growth Des. 2014, 14, 7, 3576–3586 by Proserpio and Blatov). Please concern this in the introduction (line 35).
2. Do the authors think refs 7 and 8 relevant to what was described in lines 36 and 37. I try to read refs 7 and 8 but it seems to me not relevant. Both U-MOF and ljh COF are much more complex than diamond COFs.

3. Given different bond types (H-bond, van der Waals) can be considered, statement in lines 39 and 40 is not 100% true.

4. To me the complexity of topology might not need to require non-covalent interaction, which is true in many MOF cases (see Eddaoudi's works early 2000s and other Fe-MOFs and/or Bi-MOFs). This holds true in 3D COFs. Then the argument is 'what is the definition of complexity?' Diamond is one of the most simple topologies in chemistry. What the authors described here is the complexity in the crystal structure of COF-305 but not the topological hierarchy/complexity (different levels can be broken down). That is why I think there is a 'concept swapping' existing in this work.

5. "For generating complexity, the key is to prevent the COF building blocks to pack into simple structures...": I am not sure I understand this but I think complexity of a topology depends on many factors. One of them is types of vertices and edges. If we introduce more than 1 kind of vertex and 1 kind of edge, possibilities to form many types of topologies will increase.

6. Based on these above reasons, I don't value the supplement of design proposed by the authors. It is just a case study of diamond structure and cannot be applied to other structures.

The synthesis and characterizations are straightforward and I have no concerns about them. Given the lack of novelty and 'equivocation' issue in correlating the crystal structure and topology, I think this MS needs to be massively revised before sending to another more specific journal.

Reviewer #3 (Remarks to the Author):

In this manuscript, Zhao and co-workers report synthesis and single-crystal structure of a new COF featuring the largest unit cell and asymmetric unit among known COFs. This exciting result is attributed to the molecular design of the pre-assembly of the ortho-dimethoxy topic linker and the interconnection of the tetratomic organic linker. All the experiments have been well conducted with high standard, and the presentation of results are well organized and in detail. More importantly, this work represents a millstone linking the film of supramolecular chemistry and reticular chemistry. Considering the novelty and significance, I am delighted to recommend it for publication with minor revision with the consideration as follows:

1. The authors should discuss more background on supramolecular chemistry in the introduction, especially on existing concepts of molecular pre-assembly. This would strengthen their claims on "building block packing adaptation."
2. Is similar structural complexity also known in MOFs? If so, it should also be discussed.
3. Could the authors discuss the porosities of COF-304 and COF-305?

4. Please check the latest literature on COF single-crystal structures and make comprehensive comparison with the unit cell and size of as-symmetry.
5. The structure of COF-300 should be given at the first mention to facilitate the reader's understanding.
6. Are there any similar phenomena to support the statement "These struts are all in "C" configurations, which is likely due to steric hinderance effect of the methoxy group."

** See Nature Portfolio's author and referees' website at www.nature.com/authors for information about policies, services and author benefits.

Author Responses to Reviewer#1's Comments:

The manuscript reports the exceptional complexity in COFs, resembling the hierarchical structure of DNAs. The unprecedented diversity of conformations of COFs observed by X-ray single-crystal structures with atomic precision have been elaborated in great details in both molecular geometry and energy aspects. The molecular insight revealed here is very inspiring for the delicate design of COF structures with the consideration of linker-linker interactions for controlling their spatially arrangement of building blocks. Such results are out of expectation but can be well rationale, manifesting the power of serendipity in reticular chemistry. The experiments have been well-conducted and the manuscript is well-written. Therefore, I recommend it for the publication in Nature Communications after minor revision. Several comments are provided as follows:

Here are my comments:

We appreciate the reviewer's positive comment for our work.

1. The authors have illustrated the complementary pairs of the DMPA motifs. It will be very interesting to know whether the formation of such complexity arise from the pre-assembly of monomer in reactant.

Response: This is a very important question and we have tried several methods to directly study the pre-assembly of the DMPA and its imine condensation product with aniline. We have tried to measure the NMR spectra of both molecules at different concentration, hoping to see concentration dependent chemical shifts as sign for pre-assembly. However, we could not find any chemical shift that is concentration dependent. Thus, we would speculate that either there is no strong tendency for the monomers to pre-assemble, or the formation of pre-assembled structures does not induce significant change in chemical shift. We would keep studying pre-assembly of molecular building blocks in this and other new COFs.

2. The interpenetration number of COF-305 is only 5-fold, while the COF-304 has 7-fold similar in the case of COF-300. Is it possible to explain how the methoxyl group make an impact in the interpenetration and the translational direction?

Response: The degree of interpenetration mostly depends on the number of networks that can fit in one diamondoid cage, which is essentially correlate to extent of cage elongation and ultimately related to several key bond angles (fig. R1). These bond angles are affected by the steric of the methoxy side groups and the preferred packing mode of the molecular building blocks. As is shown in fig. R1, the diamond cages of COF-304 and COF-300 are more elongated, which allow 7-fold interpenetration. For COF-305, the interaction of methoxy group between adjacent diamond cages result in less elongation at the direction of interpenetration and thus give rise to 5-fold interpenetration.

Figure R1. Geometry of adamantane cages of COF-300 (left), COF-304 (middle), and COF-305 (right).

3. As we all know that organic solvents can play as structural directing agents in the formation of porous crystals. Can the author elaborate the impact of organic solvents using in COF-305 leading the such high level of complexity?

Response: We thank the reviewer for this very insightful suggestion. We have conducted the imine condensation of DMPA and TAM in various solvents and solvent mixtures. Solvent mixtures containing dioxane would give COF-305 phase whereas pure mesitylene give another phase (structure under investigation). Thus, we do believe that dioxane could play certain structure directing role in the formation of COF-305 by mediating the inter-molecular interaction between linker molecules.

Entry	Solvent	Time	Heating temperature (°C)	Modulator	Concentration of HOAc
1	1,4-dioxane	3 days	RT	Aniline 60 uL	15M

2	1,4dioxane; mesitylene	3 days	40	Aniline 20 uL	15M
3	mesitylene	3 days	40	Aniline 20 uL	15M
4	1,4-dioxane; i-BuOH	3 days	40	Aniline 20 uL	15M
5	1,4-dioxane; DCB	3 days	40	Aniline 20 uL	15M

Figure R2. PXRD patterns of COF-305 synthesized under different organic solvent.

Figure R3. SEM image of COF-305 synthesized under different organic solvents. a. 1,4-dioxane and mesitylene; b. 1,4-dioxane and *n*-BuOH; c. 1,4-dioxane and dichlorobenzene; d mesitylene.

Author Responses to Reviewer#2's Comments:

The manuscript (MS) "Encoding Ordered Complexity to Covalent Organic Frameworks" by Zhao and coworkers describes the synthesis, characterization, and structural analysis of diamond COFs (COF-304, COF-305) in detail. From crystallographic point of view, this analysis is valuable. However, the MS lacks of novelty and does not provide inspired insights into the design of new structure types in COF chemistry (please see the below concerns and questions). Therefore, I recommend the editor not publish this work in Nature Communications. A more specific journal related to crystallography is suitable for this MS.

Response: We would like to thank the reviewers for their invaluable comment on the MS and we are sorry for the misunderstanding caused by the previous version of our MS. We strongly agree with the reviewer's comment on topological complexity of molecular frameworks. However, the exact point of this paper is to show that even with very simple topology (e.g. dia), it is still possible to achieve complex COF structures by harnessing the structural directing role of non-covalent interactions. In other words, the current MS is aiming to show that under the same topology, there is a chemical space that provide plenty of room for structural tunability, and COF-305 is an example that shows the potential of such tunability. We have no intention to argue that COF-305 is more complex than MOFs and COFs with complex topology, and we believe structural design from the perspective of non-covalent interactions is orthogonal to topological structural design and thus would be applicable to COFs with other topologies.

We very much appreciate the review's appreciation of our effort in crystallography, but we truly believe that this paper is not only about crystallography, but more importantly provides general insights for synthetic design: (a) for synthetic chemists, the take away message can be very intuitive, which says adding simple functional groups to linkers can cause different packing geometries and give framework materials with very different structure; (b) for crystal chemist, the paper shows that data mining or machine learning methods based on molecular compounds structure database can provide valuable guidelines for the design of framework materials; (c) for material chemist, the paper shows that engineering molecular packing in framework materials have the potential to tune its properties while maintaining the same network connectivity, which we would also show in the following responses. Thus, we believe this paper is of general interest to the chemistry community. We have now substantially revised our manuscript to clarify these points and emphasize our key findings.

1. Topology not only concerns covalent bonds but also relates various bonds as topology describes the connectivity (see Cryst. Growth Des. 2014, 14, 4, 1938–1949 and Cryst. Growth Des. 2014, 14, 7, 3576–3586 by Proserpio and Blatov). Please concern this in the introduction (line 35).

Response: We completely agree with the reviews that the concept of topology is to describe connectivity between molecular building blocks, and the linkage between these building blocks can either be covalent or non-covalent interactions, as long as it plays structural directing roles. In COF chemistry, the connectivity between molecular building blocks is achieve by covalent bonds, thus non-covalent bonds in COFs is not directly related to their topologies. We have

revised the manuscript to narrow down the discussion to COF chemistry in the introduction, so that there will be no misleading statements. We have cited these two papers in the new paragraph before conclusion session that discusses structural design of framework materials other than COFs .

2. Do the authors think refs 7 and 8 relevant to what was described in lines 36 and 37. I try to read refs 7 and 8 but it seems to me not relevant. Both U-MOF and ljh COF are much more complex than diamond COFs.

Response: We have revised the manuscript to clarify the difference between complex topology and structural complexity with simple topology, and these two references are relocated in the revised manuscript.

“In reticular chemistry, complexity can be achieved by topological design, where specific connectivity between molecular building blocks can direct them to form complex and ordered structures in mathematically predictable ways⁷⁻⁹.”

3. Given different bond types (H-bond, van der Waals) can be considered, statement in lines 39 and 40 is not 100% true.

Response: We agree that this statement can be misleading in different contexts, and we have now removed the statement in the re-written manuscript.

4. To me the complexity of topology might not need to require non-covalent interaction, which is true in many MOF cases (see Eddaoudi's works early 2000s and other Fe-MOFs and/or Bi-MOFs). This holds true in 3D COFs. Then the argument is 'what is the definition of complexity?' Diamond is one of the most simple topologies in chemistry. What the authors described here is the complexity in the crystal structure of COF-305 but not the topological hierarchy/complexity (different levels can be broken down). That is why I think there is a 'concept swapping' existing in this work.

Response: we completely agree with the reviewer that complexity of topology does not require non-covalent interactions. According to the principle of reticular design, achieving complex topology for COFs and MOFs almost only concerns the covalent or metal-ligand bond. However, this is not related to the key point of the current paper, which concerns the complexity of frameworks with the same topology and series of building blocks. In the context of the current paper, we define complexity as the number of non-equivalent sites in the framework that can be used to encode chemical heterogeneity and sequence. Thus, we break down the level of complexity in a synthetically intuitive manner: the first level concerns topology, which refers to the connectivity of building blocks determined by their vertices and edges; the second level concerns complexity due to different molecular packing in frameworks with the pre-determined topology. In this paper, we find that tuning non-covalent interactions could increase the complexity on the second level, which is directly related to the size of crystallographic asymmetric unit and thus the number of equivalent sites. On the other hand, adapting complex topology can also provide non-equivalent sites, as exemplified in the Bi-MOFs. However, this

approach is more difficult in COF chemistry mostly due to the crystallization challenge, where there have been very few cases of single crystalline COFs that have complex topology. Consequently, we believe our approach to increase structural complexity in COFs with simple topology is valuable in COF chemistry.

5. "For generating complexity, the key is to prevent the COF building blocks to pack into simple structures...": I am not sure I understand this but I think complexity of a topology depends on many factors. One of them is types of vertices and edges. If we introduce more than 1 kind of vertex and 1 kind of edge, possibilities to form many types of topologies will increase.

Response: We certainly agree with the reviewer that complexity of a topology would depend on many factors, including the type and vertices and edges. However, as is stated in the response for question 4, this is not really the focus of this paper. We also agree with the reviewer that introducing more than 1 kind of vertex and edge can certainly give interesting new new topologies, however it would be an approach orthogonal to the design strategy reported in this paper. It is worth noting that there has been only more than a dozen of COFs made into single crystals (Supplementary Table 4) and increasing the types of vertex and edges of the linkers would almost certainly makes crystallization more challenging.

6. Based on these above reasons, I don't value the supplement of design proposed by the authors. It is just a case study of diamond structure and cannot be applied to other structures.

Response: As is stated in the revised MS and the responses for the above questions, we strongly believe that the COF design strategy shown in this paper is generalizable and not merely a case study. We further demonstrate the generalizability of this strategy in a 2D COF series composed of 4,4',4",4'''-(pyrene-1,3,6,8-tetrayl)tetraaniline linked with terephthalaldehyde derivatives, which is currently under investigation and unpublished (figure R4). In the series, the widely studied Py-1P COF incorporates terephthalaldehyde and the newly synthesized Py-2P COF incorporates 2,3-dimethoxyterephthalaldehyde (DMPA). From the molecular diagram, the Py-1P and Py-2P COF would have the same topology (**sql**) and connectivity, and the only difference would be the molecular packing modes directed by non-covalent interactions. The Py-2P COF has very high crystallinity, which allows structure determination by 3D electron diffraction. Comparing the structure of Py-1P and Py-2P COF, they have the same intra-layer connectivity but different inter-layer packing: the layers of Py-1P are stacked in AA mode and the layers in Py-2P has more offset that lowers its symmetry and result in a crystallographic asymmetric unit twice as large as the Py-1P COF. From a topological perspective, such difference is trivial. However, such structural difference gives very different electronic and fluorescent properties. In these two COFs, the spatial arrangement of the adjacent pyrene motif between layers determines their electronic coupling and eventually affects the optoelectronic properties, which result in different colors and fluorescence wavelength. We are still conducting in-depth study on the optoelectronic properties of the Py-2P COF and thus would not include it in this MS. However, we hope such results would convince the reviewer that our design strategy can be generalized and can have actual impact on material properties and functions.

Figure R4. The Py-1P and 2P COF showing different molecular packing modes and fluorescent properties

The synthesis and characterizations are straightforward and I have no concerns about them. Given the lack of novelty and 'equivocation' issue in correlating the crystal structure and topology, I think this MS needs to be massively revised before sending to another more specific journal.

We would like to thank the reviewer again for valuable comments. From the responses above and the revised manuscript, we hope the reviewer could now appreciate the general interest of our MS and support our publication in nature communications.

Author Responses to Reviewer#3's Comments:

In this manuscript, Zhao and co-workers report synthesis and single-crystal structure of a new COF featuring the largest unit cell and asymmetric unit among known COFs. This exciting result is attributed to the molecular design of the pre-assembly of the ortho-dimethoxy topic linker and the interconnection of the tetratomic organic linker. All the experiments have been well conducted with high standard, and the presentation of results are well organized and in detail. More importantly, this work represents a millstone linking the film of supramolecular chemistry and reticular chemistry. Considering the novelty and significance, I am delighted to recommend it for publication with minor revision with the consideration as follows:

Response: We appreciate the reviewer's positive appraisal of our work.

1. The authors should discuss more background on supramolecular chemistry in the introduction, especially on existing concepts of molecular pre-assembly. This would strengthen their claims on "building block packing adaptation."

Response: We thank the reviewer for the valuable suggestions. We have now rewritten the introduction to provide better background and motivation associated with supramolecular chemistry for COF chemistry.

"Subsequently, proper molecular building blocks need to be selected to harness the degree of freedom in dia COFs and achieve structural complexity. In this regard, supramolecular chemistry provides essential toolbox to identify the molecular motifs where the non-covalent interactions between them can play structural directing roles. It is worth noting that supramolecular chemistry considerations such as molecular pre-assembly has been known to help increase the crystallinity of 2D COFs¹⁷⁻²². For the known examples of COF-300 series, the TAM molecules strongly favours a simple and robust herringbone packing that can be viewed as supramolecular synthon responsible for their relatively simple structure. Thus, to pursue structural complexity, the TAM packing would need to be altered by introducing supramolecular interactions associated with the aldehyde building blocks."

2. Is similar structural complexity also known in MOFs? If so, it should also be discussed.

Response: We have added a paragraph to discussed similar concept and complexity in MOFs and other molecular frameworks. For MOFs, structural tunability and flexibility associated with molecular building block conformation diversity is more widely observed. However, the conformation isomorphism for MOFs is almost always coupled with metal-coordination sphere distortion and thus much more difficult to predict or rationalize with supramolecular chemistry.

"We believe the scientific implications of our findings on COF-304 and 305 is not limited to COF chemistry and could be generalized to other framework materials designed by reticular chemistry. In general, reticular design employs strong and directional interactions to link up molecular building blocks in mathematically predictable ways with well-defined connectivity. These directional and dominant interactions can be viewed as the primary structure-directing

interaction for the corresponding frameworks: covalent bonds for COFs, metal-to-ligand bonds for MOFs, hydrogen bonds for hydrogen-bonded frameworks (HOFs), and so on. For each framework category, the primary interaction would almost always leave degree-of-freedom for weaker and less directional interactions to play secondary structural directing roles and give structural diversity, complexity and tunability. For example, such structural tunability through secondary structural directing interactions can also be found in MOF chemistry, where interpenetration and conformational isomers can be obtained by tuning non-covalent interactions such as solvent templating effect²⁶. Our findings in COF-305 has shown that systematically understanding and tuning secondary structural directing interactions could lead to profoundly different structures even for framework materials that seem to be very simple from the perspective of primary interactions and topological connectivity.”

3. Could the authors discuss the porosities of COF-304 and COF-305?

Response: According to the suggestion of the reviewer, we have discussed the porosity of COF-304 and 305 in the main text. The porosity of COF-304 cannot be experimentally determined due to the presence of impurity phases in its single crystal synthesis.

“The theoretical pore volume of COF-305 and COF-304 are calculated using a probe radius of 1 Å, which are 1.06 cm³g⁻¹ and 0.512 cm³g⁻¹, respectively. The pore volume of COF-305 is measured by CO₂ uptake at 195 K to be 0.91 cm³g⁻¹, which is largely consistent with the theoretical result considering the framework deformation during activation process.”

4. Please check the latest literature on COF single-crystal structures and make comprehensive comparison with the unit cell and size of as-symmetry.

Response: We have summarized the structure of latest reported single-crystal COFs which elucidated by *ab* initial method (single crystal X-ray diffraction or electron diffraction that can directly give atomic resolution without modelling). We have listed the unit cell and the number of crystallography as-symmetry unit. The table is also added to the supplementary information.

Name	topology	Unit cell	Number of as-symmetry unit
COF-904 ⁸	Interdigitating hcb	$a = 25.134 \text{ \AA}, b = 9.659 \text{ \AA}, c = 18.882 \text{ \AA},$ and $\beta = 116.48^\circ, V = 11980.4(2) \text{ \AA}^3$	1 TFB and 1.5 TMPDA
T-2DP ⁹	hcb (ABC)	$a = 26.6621(3) \text{ \AA}, b = 26.6621(3) \text{ \AA}, c = 16.8900(2) \text{ \AA}, \gamma = 120^\circ, V = 10398.0(3) \text{ \AA}^3$	1/3 M1
HCP ¹⁰	helical	$a = 15.2487(11) \text{ \AA}, b = 19.3901(11) \text{ \AA}$ and $c = 19.4093(12) \text{ \AA}, V = 5738.5(6) \text{ \AA}^3$	1 HHTP
LZU-111 ¹¹	lon-b-c3	$a = b = 20.17(3) \text{ \AA}, c = 34.0(3) \text{ \AA}, V = 11980.4(2) \text{ \AA}^3$	1 TAM and 1 TFS
Py-1P-COF ¹²	sql (AA-stack)	$a = 3.93 \text{ \AA}, b = 23.39 \text{ \AA}, c = 23.54 \text{ \AA}, \alpha = 84.5^\circ, \beta = 87.1^\circ, \gamma = 87.1^\circ, V = 2148.30 \text{ \AA}^3$	2 TPA and 1 TAPy

BP-COF-6 ¹³		$a = 5.4728(16), b = 4.3774(12), c = 14.285(4), \beta = 95.402(7)^\circ, V = 340.71(16) \text{ \AA}^3$	1 BPA-1
UTSB-20-qtz ¹⁴	qtz	$a = b = 30.935(2) \text{ \AA}, c = 40.760(8) \text{ \AA}, \alpha = \beta = 90^\circ, \gamma = 120^\circ, V = 33781(8) \text{ \AA}^3$	3/2 TAPB (or TFTB)
USTB-5 ¹⁵		$a = b = 17.9558(6), c = 19.8313(9) \text{ \AA}, \alpha = \beta = 90^\circ$ and $\gamma = 120^\circ, V = 5537.2(4) \text{ \AA}^3$	1/3 TAPB (or TFPB)
USTB-5r ¹⁵		$a = b = 18.2124(15), c = 17.975(2) \text{ \AA}, \alpha = \beta = 90^\circ$ and $\gamma = 120^\circ, V = 5163.3(10) \text{ \AA}^3$	1/3 TAPB (or TFPB)
USTB-5o ¹⁵		$a = b = 31.0280(3) \text{ \AA}, c = 39.8735(12) \text{ \AA}, \alpha = \beta = 90^\circ$ and $\gamma = 120^\circ, V = 33244.7(12) \text{ \AA}^3$	3/4 TAPB and 3/4 TFPB
mCOF-Ag ¹⁶	dia	$a = 15.83 \text{ \AA}, b = 29.97 \text{ \AA}, c = 10.69 \text{ \AA},$ and $\beta = 123.96^\circ, V = 4205.57 \text{ \AA}^3$	1 linker
UTSB-20-dia ¹⁴		$a = 23.690(5) \text{ \AA}, b = 24.980(5) \text{ \AA}, c = 27.260(6) \text{ \AA}, \alpha = 107.24(3)^\circ, \beta = 107.29(3)^\circ, \gamma = 106.56(2)^\circ, V = 5209.6(10) \text{ \AA}^3$	2 TAPB and 2 TFTB
COF-300 ¹¹		$a = b = 26.2260(18) \text{ \AA}, c = 7.5743(10) \text{ \AA}, V = 5209.6(10) \text{ \AA}^3$	1/4 TAM and 1/2 TPA
LZU-79 ¹¹		$a = b = 27.838(2) \text{ \AA}, c = 7.5132(12) \text{ \AA}, V = 5822.4(3) \text{ \AA}^3$	1/4 TAM and 1/2 BFBZ
COF-303 ¹¹		$a = b = 26.47(3) \text{ \AA}, c = 7.449(9) \text{ \AA}, V = 5220(13) \text{ \AA}^3$	1/4 TFM and 1/2 PDA
COF-320 ¹⁷		$a = 23.360(3) \text{ \AA}, c = 8.4300(17) \text{ \AA}, V = 4600.2(16) \text{ \AA}^3$	1/4 TAM and 1/2
SYSU-9 ¹⁷		$a = 26.461(4) \text{ \AA}, c = 7.4600(15) \text{ \AA}, V = 5223.6(18) \text{ \AA}^3$	1/4 TAM and 1 PDD
COF-301-s ¹⁷		$a = 26.434(4) \text{ \AA}, c = 7.5876(15) \text{ \AA}, V = 5302.0(18) \text{ \AA}^3$	1/4 TAM and 1/2 DHPA
COF-305		$a = 47.077(9) \text{ \AA}, b = 67.629(14) \text{ \AA}, c = 42.547(9) \text{ \AA},$	4.5 TAM and 9 DMPA
COF-304		$a = b = 26.210(6) \text{ \AA}, c = 7.582(3) \text{ \AA}$	1 TAM and 1/2 DHPA

5. The structure of COF-300 should be given at the first mention to facilitate the reader's understanding.

Response: According to reviewer's suggestion, the COF-300 structure is now described at the first mention.

*"In this report, we demonstrate such strategy in the chemical space of COF-300 series, which are COFs with **dia** topology formed by linking tetrakis(4-aminophenyl)methane (TAM) and terephthalaldehyde derivatives."*

6. Are there any similar phenomena to support the statement "These struts are all in "C" configurations, which is likely due to steric hinderance effect of the methoxy group."

Response: In the literature for COFs, there are no direct observation similar to this phenomena. Recently, we have synthesized a 2D COF that is solved with atomic level precision by 3D electron diffraction, which has the methoxy groups oriented towards different side of the imine bond and also giving “C” shape configuration.

Figure R5. Structure of Py-2P-COF, in which the imine bond oriented toward the different side of methoxy group.

REVIEWERS' COMMENTS

Reviewer #1 (Remarks to the Author):

I am OK with the revision and recomend the publication.

Reviewer #3 (Remarks to the Author):

This referee is satisfied with the revised version. It can now be accepted for publication.

[Note from the Editor: Reviewer #3 was asked to look also over the response given to Reviewer #2]

In my opinion, the authors have properly addressed the technical concerns of reviewer #2.